# The relationship between physical education teachers' competence support and middle school students' participation in sports: A chain mediation model of perceived competence and exercise persistence

Yubo Liu, Jianhua Yan*, Jing Li

School of Physical Education and Sport, Henan University, Kaifeng City, Henan Province, China

* 10180106@vip.henu.edu.cn

**Data Availability Statement:** All Data files are available from the DRYAD database (DOI:https://doi.org/10.5061/dryad.brv15dvkd)and uploaded as a Supporting information file.

## Abstract

This study explores the relationship between physical education teachers' support and middle school students' participation in sports. It also clarifies this relationship's mediating roles of perceived competence and exercise persistence. A questionnaire survey involved 879 Chinese middle school students, consisting of 434 males and 445 females. The results indicate a significant positive correlation between teachers' competence support and students' participation in sports ($p < 0.01$). Perceived competence and exercise persistence as important mediators between teachers' competence support and students' involvement in sports, with the mediation effect comprising independent mediation by perceived competence and exercise persistence and a chain mediation effect involving both. The effect value was 0.156, with contributions of 60.4%, 43.6%, and 73.1% to the total mediating effect, respectively. In physical education courses, teachers should enhance their competence to support students, increase students' interest in learning, and promote the development of sports participation. By understanding the predictive roles of teacher support, perceived competence, and exercise persistence on sports participation, strategies can be developed to better enhance students' levels of participation in physical activities, thereby improving their beliefs about physical health and their confidence in exercising.

## Introduction

In recent years, insufficient physical activity has emerged as a significant issue impacting the physical and mental health of adolescents [1]. Despite substantial evidence demonstrating that regular participation in physical activities offers various health benefits [2], a considerable number of middle school students fail to engage in adequate physical exercise to realize these advantages. Additionally, research on the factors influencing physical activity participation among middle school students remains relatively limited.

**Funding:** The author(s) received no specific funding for this work.

**Competing interests:** The authors have declared that no competing interests exist.

In light of this situation, there is growing attention on strategies to promote student participation in physical activities and enhance their foundational motor skills. Furthermore, the academic community acknowledges the crucial role of teacher support in influencing students' physical activity behaviors. Existing research suggests that teachers, as the direct transmitters of knowledge to adolescents, play an important role in the process of student growth. Students' perceived teacher support is an important variable that has an impact on individual psychological factors such as goal orientation and achievement motivation. Positive interactions between teachers and students play a positive role on students' cognition, emotions and behavior [3, 4]. Consequently, this study aims to investigate both the relationship and internal mechanisms between physical education teachers' support and middle school students' participation in physical activities.

Throughout the physical education curriculum, fulfilling students' psychological needs and boosting their intrinsic motivation to participate in physical activities can enhance their interest in engaging in these activities [5], which is also conducive to the development of exercise habits [6]. Nevertheless, much of the existing literature on student physical participation tends to concentrate on the effects of isolated factors. Although the support provided by teachers helped some students to increase their interest in sport, the impact of physical education teachers on student participation in sport in terms of student competence [7, 8]. The influences on students' commitment to physical activities are multifaceted, and there is comparatively little research examining the roles of teacher support, students' self-perceived capabilities, and levels of exercise persistence. This paper examines the relationship between physical education teachers' support and middle school students' engagement in physical activities, specifically exploring the mediating roles of perceived competence and exercise persistence. This investigation aims to elucidate the mechanisms through which various influences affect students' participation in physical activities, ultimately providing valuable insights for enhancing student engagement and promoting the healthy development of adolescents' physical and mental well-being.

## Theoretical foundation

The psychological needs for autonomy, competence, and relatedness are central to Self-Determination Theory (SDT) and are essential for understanding the relationship between teaching behaviors and student motivation types [9, 10]. According to SDT, when the external environment fulfills an individual's psychological needs in three areas—autonomy [11], competence [12], and relatedness [10]—individuals are motivated. This motivation facilitates the internalization of external motivation into autonomous behavioral motivation and enhances well-being, thereby increasing the persistence of individuals' efforts to engage in activities [10]. It is also considered to promote optimal functioning [13]. Furthermore, the more students' needs are satisfied, the higher their motivation levels [14]. Autonomous motivation encompasses voluntary reasons for engaging in the curriculum, such as recognizing the value of an activity or finding it enjoyable and challenging [11]. Consequently, students with high learning motivation are likely to exert greater effort in the classroom [15]. A substantial body of research based on SDT [11] indicates that a supportive teaching style effectively optimizes teaching, as competence-supportive teaching styles nurture students' fundamental psychological needs for autonomy, competence, and relatedness [16, 17]. Therefore, SDT provides a robust theoretical foundation for understanding the positive effects of teacher support.

## Literature review and research hypotheses

**The role of physical education teacher competence support in student sports participation.** Self-determination theory suggests that when the external environment satisfies an

individual's psychological needs for autonomy, competence, and relatedness, the individual becomes motivated to sustain their efforts in a particular activity [18]. Teacher support, as a crucial component of social support, serves as a key environmental factor that significantly influences the development of students' abilities. Teacher competence support pertains to establishing a well-defined structure within the PE learning environment. This is primarily accomplished by PE teachers conveying realistic expectations to students, rationalizing the rules they implement, offering constructive feedback, and tailoring physical activities to students' abilities and progress levels [8]. From the teacher's perspective, there is a tendency to focus on promoting students' independent thinking [19]. Teachers who provide autonomy support actively seek to identify, develop, and foster students' interests through strategies such as soliciting students' interests and viewpoints and using invitational language [20]. Chatzisarantis and Hagger conducted a large-scale intervention study that confirmed the positive impact of teacher support on students' extracurricular physical activity participation. Compared to students under neutral teacher support conditions, students under teacher support conditions demonstrate a stronger willingness to exercise during leisure time and participate more frequently in leisure physical activities [21]. This is because students who perceive teacher support are more likely to have their needs met, which stimulates their intrinsic motivation for sports [14]. This intrinsic motivation has a positive impact on students' sense of well-being in their participation in sports, leading to greater enjoyment of physical activities [7]. Not only in physical education but the teaching behaviors of physical education teachers and their attention to students are related to students' learning motivation and engagement in the classroom. Research has demonstrated that multiple dimensions of teacher support positively predict physical activity participation; however, there is a lack of further exploration regarding the reciprocal influence between teachers' competence support and students' physical activity participation. Thus, Hypothesis 1 is proposed: Physical education teachers' competence support can positively predict middle school students' sports participation.

**The role of student perceived competence in teacher competence support and sports participation.** Existing research has shown that perceived competence has become a key motivational determinant for students' participation in physical activities and is considered the most consistent modifiable correlate of physical activity among children and adolescents. Perceived competence can be understood as an individual's belief in their abilities across various domains of achievement. When explaining the internal mechanisms of physical education teachers' competence to support sports participation, the impact of students' perceived competence on external factors is first considered. Self-determination theory describes the relationship between students' perceived abilities and the support for internal and external motivations [14]. Generally, students with higher perceived abilities tend to participate more actively in sports activities, while students with lower perceived abilities may withdraw from sports participation due to peer ridicule or team exclusion [22]. Furthermore, students with higher sports perception abilities are more likely to engage in physical exercise during school and leisure time [23, 24]. In terms of physical activities, competence support provided by teachers, parents, coaches, and school administrators promotes students' self-motivation and improvement, which aligns with the psychological processes in self-determination theory [25]. When teachers provide competence support in physical education, students may develop teamwork, goal-setting, and leadership skills as their needs related to abilities are met [26]. Previous research has investigated how students' perceived competence impacts the extent to which they are influenced by external factors and their improvement in physical skills. However, the role of perceived competence within the internal mechanisms of teacher support and student physical activity participation has not been clearly defined, and few studies have examined the mediating effect of students' perceived competence between teacher support and

physical activity participation. Therefore, Hypothesis 2 is proposed: Student perceived competence mediates the relationship between physical education teachers' competence to support and middle school students' sports participation.

**The role of exercise persistence in the relationship between physical education teacher competence support and sports participation.** Sociologists widely agree that participation behavior is a state of persistence within an individual's behavioral process [27]. When Scanlan et al. introduced the concept of persistent behavior into the field of sports, they considered it as a psychological drive that promotes continued participation [28]. Persistence is defined as a continued investment in learning when obstacles are encountered [29]. Ahn et al. examined the relationship between college students' self-regulation and their willingness to participate in and persist with sports, demonstrating that, when controlling for intrinsic motivation and positive emotions, self-regulation competence is significantly related to the level of sports participation and the willingness to persist [30]. Previous researchers have identified persistence as an outcome variable of achievement goals in the study of motivation in sports. Additionally, many researchers have examined the positive predictive role of perceived competence on academic persistence. Regarding the predictive factors of exercise persistence for sports participation, Viira et al. indicated through their research that students with more experienced backgrounds in sports, and therefore greater physical competence, are more likely to frequently participate in physical activities compared to students with limited exercise experience [31]. Although the academic community has confirmed that exercise persistence can predict sports participation [32] and that activity levels during childhood and adolescence can predict physical activity levels in adulthood [33]. However, few studies investigate exercise persistence as an endogenous variable in physical activity participation. Furthermore, it remains to be explored whether the influence of exercise persistence on physical activity participation persists under the intervention of teacher competence support. Therefore, Hypothesis 3 is proposed: Exercise persistence mediates the relationship between physical education teacher competence support and middle school students' sports participation.

## Overall hypothetical model

In summary, although the support provided by teachers can enhance the interest in physical activity among some students, the impact of physical education teachers' support for students' competence. Therefore, the purpose of this study is to examine the influence mechanism of junior high school physical education teachers' competence support on students' physical activity participation, as well as the mediating roles of perceived competence and exercise persistence. This examination aims to facilitate the implementation of effective physical education programs, thereby increasing participation in physical activities and promoting the physical and mental health of junior high school students. Integrating Hypotheses 1 to 3, it is considered that there may exist a chain of pathways described as "teacher competence support → perceived competence → exercise persistence → sports participation." The internal mechanism by which physical education teacher competence support affects student sports participation may be that through competence support, teachers enhance students' perceived abilities, allowing students to engage in sports under external influences, thereby improving their level of exercise persistence and increasing their participation in sports activities. Accordingly, Hypothesis 4 is proposed: perceived competence and exercise persistence serve as chain mediators in the relationship between physical education teacher competence support and middle school students' sports participation. The hypothetical model is illustrated in Fig 1.

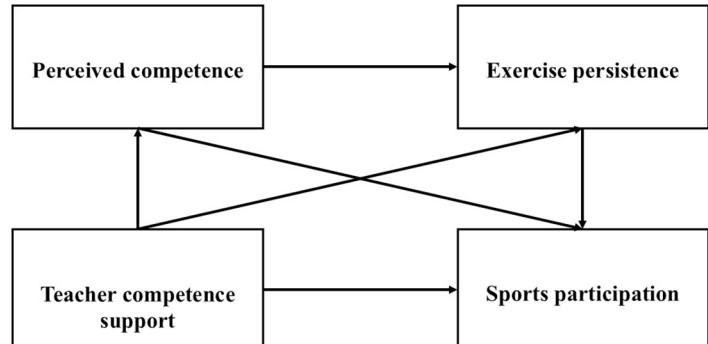

**Fig 1. Chain mediating model of teacher competence support and student exercise participation.**

## Methodology

### Participants

This study utilized a stratified cluster sampling method to conduct a survey of 1,400 students from ten middle schools in Kaifeng City, from March 10 to April 10, 2024. The survey questionnaires were distributed and collected on-site, resulting in 1,400 issued questionnaires and 1,120 returned. The study was approved by the Ethics Committee of Henan University on March 1, 2024 (approval number: HUSOM 2024–139). Before the survey, consent was obtained from the subject teachers, and the informed consent form was verbally read to the students, securing their support for participation. After excluding 241 questionnaires deemed invalid due to incomplete or incorrect responses, 879 questionnaires were included in the analysis, yielding a response rate of 80% and a valid response rate of 78.5%. The sample included 434 male students and 445 female students, with ages ranging from 12 to 17 years (M = 13.53, SD = 1.04). The distribution of students by grade level consisted of 293 first-year students (33.3%), 296 second-year students (33.7%), and 290 third-year students (33.0%).

### Instruments

**(1) Students' perceived physical education teacher competence support scale.** Students' perceived physical education teacher competence support scale was adapted from Reeve et al. [34] and Standage [35], and the Chinese version of the "Physical Classroom Needs Support Scale" developed by Yin Long was used to measure students' perceptions of teacher competence support [36]. This scale includes four items, such as "We feel that our physical education teacher provides us with many choices in class." The questionnaire utilized a 7-point Likert scale, where higher total scores indicate a greater level of perceived support. The validation results show that the confirmatory factor analysis (CFA) produced $x^2 = 328.655$, $x^2/df = 4.108$, GFI = 0.962, NFI = 0.962, IFI = 0.971, TLI = 0.965, CFI = 0.971, RMR = 0.035, RMSEA = 0.053. Reliability testing indicated a Cronbach's $\alpha = 0.814$ and a Guttman split-half coefficient of 0.893.

**(2) Perceived competence scale.** Perceived competence scales quoted from Guan [22], including statements such as "I am confident that I can master the skills taught in class." All items used a 5-point Likert scale, where higher total scores represent a greater level of perceived competence. Validation results indicated that the confirmatory factor analysis (CFA) resulted in $x^2 = 3.52$, $x^2/df = 1.76$, GFI = 0.99, NFI = 0.962, IFI = 0.921, TLI = 0.925,

CFI = 0.99, RMR = 0.02, RMSEA = 0.053. Reliability testing showed a Cronbach's α = 0.805 and a Guttman split-half coefficient of 0.823.

**(3) Exercise persistence scale.**   Exercise persistence scale were adapted from Guan [29], which includes statements such as "When I encounter difficulties in certain skills, I practice again." All items used a 5-point Likert scale, where higher total scores indicate a greater level of exercise persistence. Validation results indicated that the confirmatory factor analysis (CFA) produced $x^2$ = 32.655, $x^2$/df = 5.205, GFI = 0.932, NFI = 0.922, IFI = 0.921, TLI = 0.965, CFI = 0.971, RMR = 0.035, RMSEA = 0.043. Reliability testing indicated a Cronbach's α = 0.9 and a Guttman split-half coefficient of 0.832.

**(4) Physical activity rating scale-3, PARS-3.**   Physical activity rating scale-3 revised by Liang Qingde [37] was used to examine levels of physical exercise in terms of intensity, frequency, and duration. The formula for calculating physical exercise volume was as follows: Physical Exercise Volume = Intensity × (Duration—1) × Frequency, with each dimension scored from 1 to 5. The standards for physical exercise levels were categorized as: low exercise volume ≤ 19 points, moderate exercise volume 20–42 points, and high exercise volume ≥ 43 points. Validation results indicated that the confirmatory factor analysis (CFA) produced $x^2$ = 37.655, $x^2$/df = 4.105, GFI = 0.912, NFI = 0.902, IFI = 0.821, TLI = 0.921, CFI = 0.911, RMR = 0.045, RMSEA = 0.033. Reliability testing indicated a Cronbach's α = 0.64 and a Guttman split-half coefficient of 0.876.

## Statistical methods

Utilizing SPSS 26.0 for exploratory factor analysis of each scale. Using the PROCESS macro to analyze the data. The Harman single-factor test was conducted to prevent common method bias. To test the significance of the chain mediation model, model 6 in the PROCESS macro was used [38]. And employing AMOS 28.0 to conduct confirmatory factor analysis on the established structural equation model.

## Results

### Common method bias test

This study utilized Harman's single-factor test, incorporating all items related to physical education teacher competence support, student perceived competence, exercise persistence, and exercise participation into exploratory factor analysis. Four factors were extracted with eigenvalues greater than 1, with the first factor explaining 33.49% of the variance, which is below the critical threshold of 40% in statistical analysis. Therefore, it can be inferred that this study does not suffer from significant common method bias issues.

### Descriptive statistics and correlation analysis

The results of the Pearson bivariate correlation analysis among the variables indicate that the correlation matrix among the variables is significant (Table 1).

Among them, students' perceived teacher support is significantly positively correlated with perceived competence (r = 0.370, P<0.001), students' perceived teacher support is significantly positively correlated with exercise persistence (r = 0.440, P<0.01), students 'perceived teacher support is significantly positively correlated with exercise participation (r = 0.118, P<0.01), perceived competence is significantly positively correlated with exercise persistence (r = 0.485, P<0.01), perceived competence is significantly positively correlated with exercise participation (r = 0.268, P<0.01), and exercise persistence is significantly positively correlated with exercise participation (r = 0.189, P<0.01). These results provide preliminary support for the hypotheses

**Table 1. Variable description and correlation analysis results.**

| | M | SD | Students' perceived teacher support | Perceived competence | Exercise persistence | Exercise participation |
|---|---|---|---|---|---|---|
| Students 'perceived teacher support | 5.65 | 1.23 | 1 | | | |
| Perceived competence | 3.62 | 0.91 | 0.370** | 1 | | |
| Exercise persistence | 3.89 | 0.86 | 0.440** | 0.485** | 1 | |
| Exercise participation | 25.41 | 22.21 | 0.118** | 0.268** | 0.189** | 1 |

Note:

***P < .001,

**P < .01,

*P < .05

regarding the direct relationships among the variables in this study and also indicate that the conditions for model construction and mediation effect testing are satisfied.

## The mediating role of perceived competence and exercise persistence between teacher competence support and exercise participation

This study considers perceived competence and exercise persistence as mediating variables between teacher competence support and exercise participation. The results show that the fit indices of the three models are all less than 5, RMSEA is all less than 0.08, and the other model fit coefficients are all greater than 0.9 (Table 2), indicating that all three models have good fit and can all serve as mediating variables to explain the relationship between students' perceived teacher support and exercise participation.

To explore the positive predictive effects of teacher competence support, perceived competence, and exercise persistence on exercise participation, hierarchical regression analysis was conducted using SPSS 24.0, with teacher competence support, perceived competence, and exercise persistence as independent variables and exercise participation as the dependent variable. The regression analysis results for the chain mediation effect model are shown in Table 3.

From Table 3, it can be seen that teacher competence support significantly predicts students' exercise participation (F = 153.833, β = 0.386, P<0.001), explaining 15.0% of the variance. This indicates that middle school students' perceptions of physical education can directly influence their exercise behavior, supporting hypothesis H1. When perceived competence is included as an independent variable in the regression equation model, the predictive effect of students' perceived teacher support on exercise participation is partially significant, with statistical differences (F = 35.528, P<0.001), explaining 75.0% of the variance, indicating that perceived competence serves as a complete mediating role between teacher competence support and exercise participation, supporting hypothesis H2. When exercise persistence is included in

**Table 2. Structural equation model fit indices.**

| Fit index | X²/df | GFI | AGFI | CFI | TLI | RMSEA |
|---|---|---|---|---|---|---|
| Fit criterion | <3 | >0.90 | >0.90 | >0.90 | >0.90 | <0.08 |
| Model 1 | 2.681 | 0.963 | 0.941 | 0.950 | 0.933 | 0.061 |
| Model 2 | 2.332 | 0.973 | 0.956 | 0.965 | 0.953 | 0.052 |
| Model 3 | 2.562 | 0.962 | 0.971 | 0.969 | 0.921 | 0.062 |

Note: Model 1 has perceived competence as the mediating model; Model 2 has exercise persistence as the mediating model; Model 3 is the chain mediation model.

**Table 3. Regression analysis of teacher competence support, perceived competence, exercise persistence, and exercise participation.**

| Variable | Exercise participation | | | | | | | | |
|---|---|---|---|---|---|---|---|---|---|
| | Model 1 | | | Model 2 | | | Model 3 | | |
| | SE | β | T | SE | β | T | SE | β | T |
| Students 'perceived teacher support | 0.031 | 0.386*** | 12.403 | 0.035 | 0.042 | 1.755 | 0.037 | 0.042** | 1.125 |
| Perceived competence | | | | 0.035 | 0.222*** | 6.927 | 0.038 | 0.222*** | 5.834 |
| Exercise persistence | | | | | | | 0.039 | 0.062** | 1.572 |
| F | 153.833*** | | | 35.528*** | | | 24.548*** | | |
| R² | 0.150 | | | 0.750 | | | 0.078 | | |

Note:
***$P < .001$,
**$P < .01$,
*$P < .05$

the regression equation model, the predictive effect of teacher competence support on exercise remains significant, showing statistical differences (F = 24.584, P<0.001), explaining 7.8% of the variance, indicating that perceived competence plays a mediating role between teacher competence support and exercise participation, thus supporting hypothesis H3.

## Analysis of the chain mediation effect of perceived competence and exercise persistence between teacher competence support and exercise participation

To examine the mediating effects of perceived competence and exercise persistence on the relationship between teacher competence support and exercise participation, a structural equation model was established using AMOS software. The results of the Bootstrap test for the mediation effect are presented in Table 4 and Fig 2.

The statistical results indicate that the mediating effects of perceived competence and exercise persistence consist of three indirect effects. The indirect effect 1 of Model 1 has a Bootstrap 95% confidence interval that does not include 0, suggesting that "perceived competence " plays a significant mediating role between "teacher competence support" and "exercise

**Table 4. Bootstrap test of mediation effect.**

| Mediation model | Mediation effect | Effect size | SE | Bootstrap 95%CI | | P | Proportion of effect |
|---|---|---|---|---|---|---|---|
| | | | | Upper limit | Lower limit | | |
| Model 1 | Direct effect | 0.062 | 0.033 | -0.007 | 0.131 | 0.796 | 39.6% |
| | Indirect effect | 0.094 | 0.016 | 0.064 | 0.129 | <0.05 | 60.4% |
| | Total effect | 0.156 | 0.033 | 0.091 | 0.222 | <0.01 | 100% |
| Model 2 | Direct effect | 0.088 | 0.037 | 0.015 | 0.161 | <0.05 | 56.4% |
| | Indirect effect | 0.068 | 0.018 | 0.033 | 0.104 | <0.01 | 43.6% |
| | Total effect | 0.156 | 0.033 | 0.091 | 0.222 | <0.01 | 100% |
| Model 3 | Direct effect | 0.042 | 0.037 | 0.091 | 0.221 | <0.01 | 26.9% |
| | Indirect effect | 0.114 | 0.022 | 0.074 | 0.159 | <0.01 | 73.1% |
| | Total effect | 0.156 | 0.033 | 0.091 | 0.222 | <0.01 | 100% |

Note: Model 1 has perceived competence as the mediating model; Model 2 has exercise persistence as the mediating model; Model 3 is the chain mediation model.

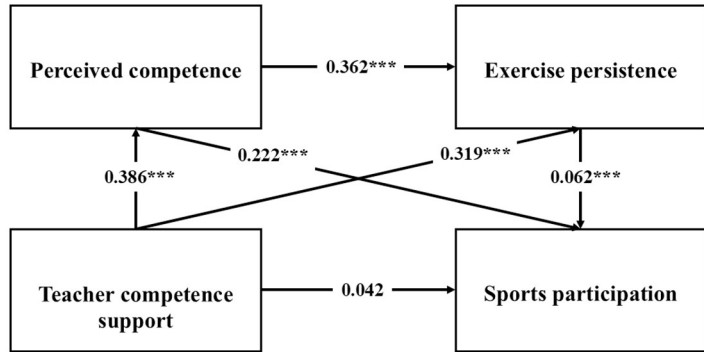

**Fig 2. Path analysis of SEM.** *** p < 0.001.

participation," with a mediating effect proportion of 60.4%. The indirect effect 2 generated by Model 2 also has a Bootstrap 95% confidence interval that does not include 0, indicating that "exercise persistence" significantly mediates the relationship between "teacher competence support" and "exercise participation," with a mediating effect proportion of 43.6%. The indirect effect 3 from the paths in Model 3 has a Bootstrap 95% confidence interval that does not include 0, demonstrating that "perceived competence " and "exercise persistence" significantly mediate the relationship between "teacher competence support" and "exercise participation," with a mediating effect proportion of 33.33%, thereby supporting hypothesis H4.

## Discussion

The study found that physical education teacher competence support can directly and significantly positively predict middle school students' exercise behavior. Specifically, as teacher competence support increases, so too does the frequency, duration, and volume of exercise among junior high school students. This finding is consistent with previous research indicating that "teachers who provide autonomy support foster a specific set of intrinsic motivational rules, thereby promoting students' exercise participation" [39], and that "teacher support has a significant predictive effect on students' intrinsic motivation for physical activities" [40]. In an environment characterized by teacher competence support, such support positively influences students' enjoyment of exercise and their participation in physical activities, thereby facilitating the implementation of effective physical education curricula and enhancing participation rates and mental health among junior high school students [7, 41].

Based on self-determination theory, the relationship between physical education teachers' competence to provide support and students' physical activities can be examined from two perspectives: competence satisfaction and intrinsic motivation. Specifically, when teachers implement effective strategies, students' psychological needs are fulfilled, thereby fostering their active participation in physical activities [25].

### Direct effect of physical education teacher competence support

According to Hypothesis 1, physical education teacher competence support can positively predict middle school students' involvement in sports. This finding is consistent with the assumptions of self-determination theory [42]. Koka A's research explored the effects of teacher cognitive support, process support, and organizational autonomy support on students' intrinsic motivation. The results indicated that the autonomy support provided by teachers significantly enhanced students' intrinsic motivation, thereby promoting their participation in sports

[40]. Through competence support, physical education teachers can significantly elevate students' levels of participation in sports, primarily reflected in several aspects: First, teacher competence support can strengthen students' self-efficacy. Bandura's self-efficacy theory posits that an individual's confidence in their abilities is a crucial factor influencing their participation behaviors [43]. Teachers who provide competence support can enhance students' perception of their capabilities by offering positive feedback and reasonable challenges, making them more willing to engage in physical activities. This positive emotional feedback not only boosts students' confidence but also enables them to become more actively involved in exercise. Additionally, surveys indicate that the competence support provided by teachers helps students better understand sports skills, reducing psychological barriers to participation. By designing exercises and competitive activities suitable for students' competence levels, physical education teachers enable students to experience moderate challenges and a sense of achievement during participation, thereby increasing their interest in sports. Furthermore, the guidance and encouragement provided by teachers during the competence support process can alleviate students' fears of failure, encouraging them to take on challenges. This environment not only instills a sense of safety in students but also significantly promotes their ongoing participation in sports.

Teachers' competence support can assist students in proactively selecting sports that match their skill levels based on their positive responses before participating in physical activities, thereby creating a more conducive learning environment. This finding emphasizes that the role of physical education teachers extends beyond merely fulfilling basic teaching duties and establishing foundational support relationships. Furthermore, teachers should strive to create a positive exercise atmosphere based on the feedback provided by competence support, enhancing students' experiences and sense of achievement during physical exercise. These research findings hold significant practical implications for teacher training and curriculum design.

In terms of teacher training, training programs should focus on enhancing teachers' understanding and application of competence support for students, helping them learn how to adjust teaching strategies according to students' individual differences and responses. This includes not only the instruction of physical skills but also psychological support and emotional management, enabling teachers to better motivate students to engage in physical activities.

Regarding curriculum design, programs should be flexible and varied, featuring a range of adjustable difficulty sports tailored to different students' skill levels and interests, allowing students to freely choose and participate in a safe and supportive environment. This personalized teaching approach will help improve students' confidence, sense of participation, and interest in sports, thus driving them to achieve deeper satisfaction in physical exercise. By implementing these concepts in teacher training and curriculum design, it is possible to effectively enhance students' performance and experiences in physical activities, thereby promoting their overall development.

## Mediating effect of student perceived competence

According to Hypothesis 2, student perceived competence mediates the relationship between physical education teacher competence support and middle school students' sports participation. This conclusion is in line with Miller's findings, which emphasize the importance of intrinsic motivation in activity participation within self-determination theory, highlighting that "intrinsic motivation is the most enduring form of motivation, and satisfying intrinsic motivation promotes personal enjoyment and self-regulation of behavior" [44]. This

theoretical foundation offers a critical perspective for understanding how students perceive teacher competence support and transform it into motivation for sports participation.

Existing research has verified a significant correlation between students' perceived competence and sports participation. For instance, assessments of participation in school sports and perceived social support show a positive correlation between high levels of perceived competence among middle school students and their engagement in sports behavior [45]. This indicates that students' perceived competence not only reflects their response to physical education teacher support but also further influences their sports participation behaviors. Throughout the process of physical education, teacher support creates a favorable external drive for middle school students. This support manifests not only in assigning tasks, conducting assessments, and providing incentives but also subtly influences students' exercise behaviors through emotional support, feedback, and guidance. As Deci and Ryan pointed out, teachers can enhance students' intrinsic motivation by providing appropriate support, thereby increasing their enthusiasm and persistence in sports participation [11]. When students feel supported by their teachers, they are more likely to translate these subjective experiences into a deeper understanding of physical activities, leading to a more positive attitude toward participation, seeking inner joy and satisfaction.

Data analysis in this study revealed a highly significant mediating effect of student perceived competence between physical education teacher competence support and sports participation. This finding further illustrates the complexity of interactions between teacher support and students' perceived competence in educational environments and how they collectively promote student involvement in sports activities. This transformational process not only highlights the close connection between physical education and cognitive development among middle school students but also emphasizes the key role of teacher support in shaping students' attitudes and experiences related to sports participation. This suggests that in the process of physical education teaching, it is essential not only to focus on the role of physical education teachers' competence support as a single factor but also to fully leverage the mediating role of students' perceived competence and other subjective factors in physical activity participation. By enhancing teacher training, systematic training programs should be developed around how to strengthen teachers' competence support, particularly in the areas of psychological and emotional support. Teachers need to learn how to effectively identify and enhance students' perceived competence in order to better motivate them to participate during instruction. Moreover, training should include strategies for creating a positive learning environment, which enables students to feel supported and valued in physical activities, thereby increasing their confidence and willingness to participate. In terms of curriculum design, educators should flexibly adjust course content to accommodate the varying abilities and perceptions of different students. Designing diverse physical activities that allow students to choose participation based on their own levels and interests will help improve their exercise experience and sense of achievement. Additionally, the curriculum should incorporate the cultivation of students' perceived competence, for instance, through reflective practice or peer feedback activities, to promote interaction and collaboration among students, enhancing their enthusiasm for participation through social support.

## Mediating effect of exercise persistence

According to Hypothesis 3, exercise persistence serves as a mediating factor between physical education teacher competence support and middle school students' involvement in sports. In physical education, a teaching style that supports students' psychological needs effectively enhances students' intrinsic motivation for exercise, resulting in increased effort and

engagement during physical education classes. This heightened engagement from students translates into higher levels of sports participation and adherence [32]. When students perceive effective support from their teachers, their energy and involvement in class significantly increase, reflected by higher levels of sports participation and sustained exercise persistence.

Consistent with our findings, Ley et al. examined the relationship between self-regulation, sports participation, and adherence intentions among college students, revealing that, after controlling for intrinsic motivation and positive emotions, self-regulation was significantly related to levels of sports participation and adherence intentions [46]. Specifically, through effective competence support, teachers can help students develop positive attitudes and confidence towards physical activity. This support includes not only technical guidance and emotional encouragement but also recognizing students' efforts. For example, our research found that when teachers provide timely positive feedback, students' exercise motivation and willingness to adhere significantly increase. Further analysis indicates that exercise persistence, as a mediating variable, has a significant impact on the relationship between teacher competence support and student sports participation. Teacher support first enhances students' intrinsic motivation for exercise, which subsequently fosters exercise persistence and ultimately leads to enhanced levels of participation.

This finding suggests that exercise persistence mediates the relationship between physical education teacher competence support and middle school students' sports participation, and by enhancing students' exercise motivation and self-efficacy, it facilitates better participation in physical activities. Curriculum content design should consider how to stimulate students' interest through a variety of physical activities and ensure that these activities can enhance students' self-confidence. Teachers can design sports of varying difficulty and types, allowing students to experience a sense of achievement through challenges while enhancing their intrinsic motivation for physical activity. It is also emphasized that teachers play a key role in educational practice; their supportive behaviors can effectively promote students' persistence in exercise, laying a foundation for long-term participation in physical activities. In subsequent training, special attention should be given to how to cultivate teachers' supporting skills, including how to effectively stimulate students' motivation for physical activity and their self-efficacy. This can be practiced through demonstrations, role-playing, and reflective teaching, enabling teachers to learn how to provide targeted support and feedback in real situations to encourage students' long-term engagement. In summary, physical education teachers should establish a supportive environment and provide positive feedback to motivate student participation, thereby promoting their future exercise behaviors.

## Chain mediating effect of student perceived competence and exercise persistence

According to Hypothesis 4, student perceived competence and exercise persistence act as chain mediating factors between physical education teacher competence support and middle school students' sports participation. This study reveals how teacher competence support enhances students' perceived competence, which in turn strengthens exercise persistence, subsequently affecting students' sports participation. This finding provides a novel perspective on how to effectively promote sports participation among middle school students. Previous research has also indicated that students with higher perceived competence are more likely to demonstrate enjoyment and persistence in physical activities [22]. Our study similarly finds that when teachers provide targeted competence support, students' perceived abilities significantly increase, thereby enhancing their participation in physical activities. This suggests that

teacher competence support encompasses not only technical guidance but also emotional support and feedback, all of which contribute to boosting students' self-perception and perceived abilities.

Building on this, we also observed that students with low levels of academic self-efficacy benefit from student-centered teaching methods. This approach enhances perceived competence by increasing students' autonomy and subsequently influencing their sports participation [47]. Teacher support also meets students' autonomy needs, resulting in increased perception of their own skills [48]. Thus, beyond focusing on the development of physical skills and fitness, physical education teachers should emphasize students' psychological adaptability in their teaching, creating an environment that promotes learning and active participation while providing multi-level support. Furthermore, this study highlights the interactive relationship between perceived competence and exercise persistence. Students with high perceived competence tend to establish stronger adherence to exercise, as they can find enjoyment in physical activity and are less likely to give up in the face of setbacks. This chain mediating model reveals the complexity of physical education teaching: teacher competence support first enhances students' perceived abilities, which consequently strengthens their exercise persistence, ultimately promoting participation in physical activities. This finding provides a theoretical basis for physical education practice and underscores the importance of aiding students in enhancing their perceived competence and exercise persistence during the teaching process.

In summary, the competence support and engagement of teachers, along with middle school students' perceived abilities and sports participation, are significant factors influencing students' engagement in physical activities. This suggests that future physical education practices should prioritize this chain mediation mechanism, encouraging teachers to implement more personalized and supportive teaching strategies to help students overcome challenges in sports and cultivate lasting exercise habits. This not only contributes to improving students' physical fitness but also promotes their psychological well-being and social adaptability, achieving the comprehensive educational goal of overall development.

## Research conclusions, implications, and limitations

### Research conclusion

The teacher competence support, student perceived competence, exercise persistence, and sports participation have significant positive correlations with one another. In the relationship between teacher competence support and student sports participation, perceived competence and exercise persistence serve as significant mediators. The mediating effect consists of both the independent mediation of perceived competence and exercise persistence, as well as the chain mediation between the two. To this end, schools can provide various capacity support training and teaching competitions for teachers to enhance their competence to support teaching. Teachers should establish a positive classroom atmosphere and provide timely and constructive feedback to enhance students' self-efficacy and motivation for physical activity, enabling students to exert more effort and optimize their perception of competence, while also recognizing the importance and interest of physical education. In the future, research can continue to be conducted on the interplay between the different factors and cultures that influence students' participation in sports, and to explore interventions between them. Curriculum design should incorporate diverse activities that integrate elements of knowledge, skills, games, and emotional attitudes to enhance adolescents' recognition of the value of physical education, stimulate their positive emotional experiences, and encourage their participation in physical exercise. Furthermore, we should explore the educational value of physical education,

integrating it with knowledge, skills, games, and emotional attitudes to enhance adolescents' recognition of the value of physical education, stimulate positive emotional experiences, and encourage students' enthusiasm for physical exercise.

## Research implications

Based on the comprehensive results and discussions, the following implications related to sports learning engagement can be summarized:

During physical education instruction, physical education teachers should not only complete basic teaching tasks and establish fundamental supportive relationships but also create a positive exercise atmosphere based on the competence feedback provided by their teaching support. This approach aims to enhance students' experiential and gainful feelings during physical exercise. It is important to not only focus on the role of teacher competence support as an individual factor but also to fully leverage subjective factors, such as students' perceived abilities, in mediating sports participation.

During the learning process, heuristic teaching methods can be employed to cultivate students' competence to reflect, revise, and analyze knowledge of movement skills. This helps establish students' self-esteem and promotes positive, independent feedback on their learning experiences. Consequently, it enables more effective teaching strategies and techniques in physical education practice.

## Research limitations

Despite this study's examination of the influence of physical education teacher competence support on middle school students' sports participation through the lens of self-determination theory, there are several limitations:

First, this study employs a cross-sectional design, which cannot directly reveal the causal relationships between variables. Therefore, when analyzing the predictive effects of various variables, we primarily rely on the support of theoretical assumptions. To more accurately explore the dynamic relationships and causal mechanisms among these variables, future research should consider adopting longitudinal study designs or experimental methods to track the effects of variables over time.

Second, the sample of this study is limited to middle school students, and the geographical scope only covers Kaifeng City, which somewhat restricts the regional representativeness of the sample and the generalizability of the research findings. To obtain more comprehensive and widespread research results, future studies should include samples from students of different age groups and regional backgrounds to enhance the external validity of the research. Furthermore, considering that cultural and educational backgrounds in different regions may have varying impacts on teacher support and student participation, comparative studies across regions would also contribute to a deeper understanding of this topic.

Finally, this study primarily focuses on the impact of teacher competence support on the learning environment, without fully exploring the effects of other environmental factors (such as peer support, family environment, and school culture) on students' participation in physical activities, perceived control, and ultimate outcomes. To gain a deeper understanding of the complex relationship between the learning environment and students' participation in physical activities, future research should broaden its scope and consider the influence of additional relevant factors. This approach will not only help elucidate the mechanisms of teacher support but also provide a more comprehensive perspective for improving students' experiences in physical activity participation.

## Supporting information

**S1 File.**
(SAV)

## Acknowledgments

We would like to thank the respondents who took part in our survey.

## Author Contributions

**Conceptualization:** Yubo Liu.

**Data curation:** Yubo Liu.

**Formal analysis:** Yubo Liu.

**Funding acquisition:** Jianhua Yan.

**Investigation:** Yubo Liu, Jianhua Yan, Jing Li.

**Methodology:** Yubo Liu, Jianhua Yan, Jing Li.

**Project administration:** Jianhua Yan.

**Software:** Yubo Liu.

**Supervision:** Jianhua Yan.

**Validation:** Yubo Liu.

**Writing – original draft:** Yubo Liu.

**Writing – review & editing:** Jianhua Yan.

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
