## [Decision Letter · Decision Letter 0]

26 Nov 2024

PONE-D-24-49663The Relationship Between Physical Education Teachers' Capability Support and Middle School Students' Participation in Sports: A Chain Mediation Model of Perceived Capability and Exercise PersistencePLOS ONE

Dear Dr. Yan,

Thank you for submitting your manuscript to PLOS ONE. After careful consideration, we feel that it has merit but does not fully meet PLOS ONE’s publication criteria as it currently stands. Therefore, we invite you to submit a revised version of the manuscript that addresses the points raised during the review process.

We look forward to receiving your revised manuscript.

Kind regards,

Henri Tilga, PhD

Academic Editor

PLOS ONE

Journal Requirements:

2. We suggest you thoroughly copyedit your manuscript for language usage, spelling, and grammar. If you do not know anyone who can help you do this, you may wish to consider employing a professional scientific editing service. The American Journal Experts (AJE) (https://www.aje.com/) is one such service that has extensive experience helping authors meet PLOS guidelines and can provide language editing, translation, manuscript formatting, and figure formatting to ensure your manuscript meets our submission guidelines. Please note that having the manuscript copyedited by AJE or any other editing services does not guarantee selection for peer review or acceptance for publication. Upon resubmission, please provide the following: ● The name of the colleague or the details of the professional service that edited your manuscript ● A copy of your manuscript showing your changes by either highlighting them or using track changes (uploaded as a *supporting information* file) ● A clean copy of the edited manuscript (uploaded as the new *manuscript* file)

3. We note that you have indicated that there are restrictions to data sharing for this study. PLOS only allows data to be available upon request if there are legal or ethical restrictions on sharing data publicly. For more information on unacceptable data access restrictions, please see http://journals.plos.org/plosone/s/data-availability#loc-unacceptable-data-access-restrictions. Before we proceed with your manuscript, please address the following prompts: a) If there are ethical or legal restrictions on sharing a de-identified data set, please explain them in detail (e.g., data contain potentially identifying or sensitive patient information, data are owned by a third-party organization, etc.) and who has imposed them (e.g., a Research Ethics Committee or Institutional Review Board, etc.). Please also provide contact information for a data access committee, ethics committee, or other institutional body to which data requests may be sent. b) If there are no restrictions, please upload the minimal anonymized data set necessary to replicate your study findings to a stable, public repository and provide us with the relevant URLs, DOIs, or accession numbers. For a list of recommended repositories, please see https://journals.plos.org/plosone/s/recommended-repositories. You also have the option of uploading the data as Supporting Information files, but we would recommend depositing data directly to a data repository if possible. We will update your Data Availability statement on your behalf to reflect the information you provide.

Reviewers' comments:

Reviewer's Responses to Questions

**Comments to the Author**

1. Is the manuscript technically sound, and do the data support the conclusions?

Reviewer #1: Yes

Reviewer #2: Yes

2. Has the statistical analysis been performed appropriately and rigorously? 

Reviewer #1: Yes

Reviewer #2: Yes

3. Have the authors made all data underlying the findings in their manuscript fully available?

Reviewer #1: Yes

Reviewer #2: Yes

4. Is the manuscript presented in an intelligible fashion and written in standard English?

Reviewer #1: No

Reviewer #2: Yes

5. Review Comments to the Author

Reviewer #1: The topic of the paper holds a certain degree of significance,and the overall design is relatively sound,employing appropriate research methodologies to address the research questions.The processing of quantitative data complies with relevant standards.However,the paper still presents the following issues:

• Pay attention to the accuracy of citations.

For example, the author said

Teacher support,an important external environmental factor widely advocated in self-determination theory,is generally considered from the perspective of students[6].

Self-determination theory has never emphasized teacher support;it has only highlighted the external environment.Teacher support is a significant part of the external environment.Therefore,when discussing this concept,it should be stated that teacher support,as an integral part of important social support or as a crucial environmental variable,plays a significant role in the development of students'capabilities.The author's current phrasing should be avoided,as there are many similar sentences throughout the text.

• The entire text must be polished,especially in terms of expression,where many parts can lead to ambiguity,as exemplified earlier.Additionally,there are traces of awkward translations from Chinese to English in several places.

• Sentences should generally not be too long.For instance,the first sentence of the abstract spans 4-5 lines,which could be broken down for clarity.

• Please carefully refer to high-level publications,such as quantitative research papers published in this journal,to revise the terminology throughout the paper.For example,the third section should be titled"Methodology,"and research tools should be referred to as"instruments"or"measures."

• Consistency is required when reporting data.For example,the chi-square and degrees of freedom ratio can be reported together,but the author uses separate expressions when reporting the KMO value corresponding to the chi-square degrees of freedom ratio for the teacher support dimension.

• The author should clearly explain the meaning of several core variables in the literature review,such as what constitutes teacher support and student ability.Additionally,"teacher support"should be referred to as"students'perceived teacher support,"as it is measured and perceived from the student's perspective.

• There should be a corresponding theoretical discussion on why the current topic is researched from a particular angle.

• The expression of the research gap is not particularly accurate,and there is a lack of effective discussion on the purpose of the research.

• Interdisciplinary research has certain guiding significance for this study;please refer to the following literature.

Liu, H., Li, X., & Y. Yan (2023) Demystifying the predictive role of students’ perceived foreign language teacher support in foreign language anxiety: the mediating role of L2 grit. Journal of Multilingual and Multicultural Development. https://doi.org/10.1080/01434632.2023.2223171.

Liu, H. & Li, X. (2023). Unravelling students’ perceived EFL teacher support. System. 115, 103048, 1-12. https://doi.org/10.1016/j.system.2023.103048.

Li, X., Duan, S., & Liu, H. (2023). Unveiling the predictive effect of students’ perceived EFL teacher support on academic achievement: The mediating role of academic buoyancy. Sustainability, 15, 10205. 1-12. https://doi.org/10.3390/su151310205

Please Note there is no relationship between my review and your citations of the above references.

Reviewer #2: Thank you to the editor for the opportunity to review this manuscript and to the authors for their diligent work on this important and timely topic. The study addresses an essential area of physical education by exploring how teacher support influences sports participation, mediated through perceived capability and exercise persistence. While the manuscript has notable strengths, I have some important comments that need to be addressed before it can be considered for publication.

INTRODUCTION

The introduction provides a solid rationale for the study by emphasizing the importance of physical education and teacher support in promoting sports participation. However, it could benefit from additional depth in certain areas:

1. The review of prior research is somewhat limited in scope, with insufficient discussion of studies from diverse cultural contexts or recent advances in the field.

2. The conceptual framework is introduced but could be better integrated with the study objectives and hypotheses.

Suggestions:

a. Broaden the literature review to include more diverse studies and recent developments in the role of teacher support in sports participation.

b. Strengthen the link between the conceptual framework and the stated hypotheses to clarify the study's novelty and theoretical contribution.

LITERATURE REVIEW AND HYPOTHESES

The hypotheses are logical and supported by previous research, but the literature review lacks depth in exploring certain mediating mechanisms, particularly the interaction between perceived capability and exercise persistence.

Suggestions:

a. Expand the discussion of the chain mediation effect, providing more theoretical grounding for how perceived capability and exercise persistence jointly influence sports participation.

b. Ensure that all hypotheses are clearly tied to specific gaps in the literature to highlight their significance.

METHODS

The methodology is well-documented, with a clear description of the sampling procedure and the validated instruments used. However:

1. The sample is geographically restricted to one city in China, which limits the generalizability of findings.

2. The use of cross-sectional data limits causal inferences, a limitation that should be acknowledged in more detail.

Suggestions:

a. Acknowledge the geographic and demographic limitations of the sample and their implications for generalizability.

b. Discuss the limitations of cross-sectional design and suggest longitudinal or experimental approaches for future research.

RESULTS

The results are presented systematically, and the statistical analyses are robust. However:

1. The reporting of results could be streamlined to avoid redundancy, particularly in describing mediation effects.

2. The inclusion of visual aids (e.g., diagrams of mediation pathways) could enhance clarity.

Suggestions:

a. Consolidate the presentation of results to eliminate repetitive descriptions.

b. Use visuals to succinctly illustrate key findings, such as the mediation pathways and model fit indices.

DISCUSSION

The discussion effectively interprets the findings but could benefit from more focus and specificity:

1. The implications of the findings for educational practice are briefly mentioned but not fully developed.

2. Some sections reiterate results rather than offering deeper insights into their significance.

Suggestions:

a. Expand on the practical implications of the findings, particularly for teacher training and curriculum design.

b. Focus on critical insights and avoid restating results already presented.

CONCLUSION

The conclusion provides a succinct summary of the findings but could emphasize actionable recommendations more strongly.

Suggestions:

a. Include specific recommendations for physical education practices, such as strategies for enhancing teacher support and fostering student engagement in sports.

b. Highlight potential areas for future research, such as examining other mediators or exploring interventions in different cultural contexts.

LIMITATIONS

The study appropriately acknowledges some limitations but could delve deeper into others:

1. The geographic restriction of the sample and the cross-sectional design are significant constraints that should be discussed more comprehensively.

2. The potential influence of unmeasured variables, such as socioeconomic status or school-level factors, is not addressed.

Suggestions:

a. Expand on the limitations related to the study design and sample characteristics.

b. Suggest directions for future research, such as incorporating longitudinal designs or exploring additional influencing factors.

This manuscript makes a valuable contribution to understanding the role of teacher support in sports participation among middle school students. Addressing the outlined suggestions, such as enriching the literature review, refining the discussion, and emphasizing practical implications, will significantly enhance the clarity, rigor, and impact of the manuscript.

Again, thank you and congratulations!

6. PLOS authors have the option to publish the peer review history of their article (what does this mean?). If published, this will include your full peer review and any attached files.

Reviewer #1: No

Reviewer #2: **Yes: **Joseph Lobo

---

## [Author Response · Author response to Decision Letter 0]

16 Dec 2024

Response to Reviewers

Thank you to the two reviewers for their valuable suggestions on improving the manuscript. We have responded to journal requirements and each comment and made all of the suggested changes. The paper was also proofread to avoid grammatical errors.

Journal Requirements:

1.Please ensure that your manuscript meets PLOS ONE's style requirements, including those for file naming.

Response 1: We revised the formatting requirements and file naming requirements of the manuscript to comply with the journal's requirements.

2. We suggest you thoroughly copyedit your manuscript for language usage, spelling, and grammar.

Response 2: We re-edited the manuscript for language use, spelling, and grammar to meet the journal's requirements.

3. Data Availability

Response 3: We uploaded the data to the journal's designated database, DRYAD, DOI: https://doi.org/10.5061/dryad.brv15dvkd. and uploaded the data as a support information file.

Reciew#1

Comment 1. Pay attention to the accuracy of citations.

For example, the author said Teacher support, an important external environmental factor widely advocated in self-determination theory, is generally considered from the perspective of students [6].

Self-determination theory has never emphasized teacher support; it has only highlighted the external environment. Teacher support is a significant part of the external environment. Therefore, when discussing this concept, it should be stated that teacher support, as an integral part of important social support or as a crucial environmental variable, plays a significant role in the development of students' capabilities. The author's current phrasing should be avoided, as there are many similar sentences throughout the text.

Response 1: We really appreciate your comments on this matter. We reorganized the relationship between teacher support and self-determination theory and revised this section in the text. Please see L102-L107.

Comment 2. The entire text must be polished, especially in terms of expression, where many parts can lead to ambiguity, as exemplified earlier. Additionally, there are traces of awkward translations from Chinese to English in several places.

Response 2: We have revised the article to address some of the problems that existed in terms of expression, as well as traces of insufficient fluency in Chinese and English translations, and we have embellished the entire article.

Comment 3. Sentences should generally not be too long. For instance, the first sentence of the abstract spans 4-5 lines, which could be broken down for clarity.

Response 3: We broke up sentences that were too long in the article to improve overall clarity. Please see L13-L19.

Comment 4. Please carefully refer to high-level publications, such as quantitative research papers published in this journal, to revise the terminology throughout the paper. For example, the third section should be titled "Methodology," and research tools should be referred to as "instruments" or "measures."

Response 4: We referenced high-level publications and revised the terminology in the article. Please see L225 and L242.

Comment 5. Consistency is required when reporting data. For example, the chi-square and degrees of freedom ratio can be reported together, but the author uses separate expressions when reporting the KMO value corresponding to the chi-square degrees of freedom ratio for the teacher support dimension.

Response 5: We modified the reporting of chi-square values and degrees of freedom ratios to ensure consistent data reporting. Please see L261-L265, L270-L274, L283-L287.

Comment 6. The author should clearly explain the meaning of several core variables in the literature review, such as what constitutes teacher support and student ability. Additionally, "teacher support" should be referred to as" students' perceived teacher support," as it is measured and perceived from the student's perspective.

Response 6: We added explanations of several core variables to the literature review and revised the section where “teacher support” should be called “students' perceived teacher support.” Please see L49, L310, L312, L314, L315, L334, L349, L357.

Comment 7. There should be a corresponding theoretical discussion on why the current topic is researched from a particular angle.

Response 7: We added a theoretical foundations section as a way to ensure a specific perspective on the research topic. Please see L77-L98.

Comment 8. The expression of the research gap is not particularly accurate, and there is a lack of effective discussion on the purpose of the research.

Response 8: We added a current research gap section to ensure effective discussion of the research hypothesis and research objectives. Please see L127-L134, L160-L167, L193-L197, L202-L211.

Comment 9. Interdisciplinary research has certain guiding significance for this study; please refer to the following literature.

Response 9: We have carefully referenced several of the papers you submitted and similar interdisciplinary research has been very enlightening for this study. We have added a number of interdisciplinary themes to our study and have made modifications to our research. For example:

[1] Reeve, J., & Jang, H. (2006). What teachers say and do to support students' autonomy during a learning activity. Journal of educational psychology, 98(1), 209.

[2] Ryan, R. M., & Deci, E. L. (2000). Self-determination theory and the facilitation of intrinsic motivation, social development, and well-being. American psychologist, 55(1), 68.

[3] Skinner, E. A., & Belmont, M. J. (1993). Motivation in the classroom: Reciprocal effects of teacher behavior and student engagement across the school year. Journal of educational psychology, 85(4), 571.

Review#2

INTRODUCTION

Comment a. Broaden the literature review to include more diverse studies and recent developments in the role of teacher support in sports participation.

Response a: We have added recent advances and diverse research on teacher support in sport participation. Please see L47-L53.For example:

[1]Ghorbani S, Nouhpisheh S, Shakki M. Gender differences in the relationship between perceived competence and physical activity in middle school students: Mediating role of enjoyment. International journal of school health. 2020;7(2):14-20.

[2] Eberline A, Judge LW, Walsh A, Hensley LD. Relationship of enjoyment, perceived competence, and cardiorespiratory fitness to physical activity levels of elementary school children. Physical Educator. 2018;75(3):394-413.

Comment b. Strengthen the link between the conceptual framework and the stated hypotheses to clarify the study's novelty and theoretical contribution.

Response b: We strengthened the link between the conceptual framework and the research hypothesis. Please see L63-L65.

LITERATURE REVIEW AND HYPOTHESES

Comment a. Expand the discussion of the chain mediation effect, providing more theoretical grounding for how perceived capability and exercise persistence jointly influence sports participation.

Response a: We extend the discussion of chain mediation effects and add a Theoretical Foundation section. Please see L77-L98,L202-L211.

Comment b. Ensure that all hypotheses are clearly tied to specific gaps in the literature to highlight their significance

Response b: We have added a description of the research gaps in the literature review section, highlighting the links between the research hypotheses and specific research gaps. Please see L127-L134, L160-L167, L182-L185,L193-L197.

METHODS

Comment a. Acknowledge the geographic and demographic limitations of the sample and their implications for generalizability.

Response a. We added geographic and demographic limitations to the sample and placed this section in the Research Limitations section. Please see L686-L713.

Comment b. Discuss the limitations of cross-sectional design and suggest longitudinal or experimental approaches for future research.

Response b: Similarly, we add a discussion of the limitations of the article's cross-sectional design and suggest a longitudinal or experimental approach for future studies. Please see L686-L713.

RESULTS

Comment a. Consolidate the presentation of results to eliminate repetitive descriptions.

Response a: We simplified the reporting of descriptive findings. Please see L358-L362, L364-L368.

Comment b. Use visuals to succinctly illustrate key findings, such as the mediation pathways and model fit indices.

Response b: We added visual tools for mediating paths and model fit indices. Please see L380.

DISCUSSION

Comment a. Expand on the practical implications of the findings, particularly for teacher training and curriculum design.

Response a: We increased the practical implications of the findings in order to facilitate an increase in the impact of this study on physical education teaching and learning. Please see L448-L475, L515-L537, 570-L584.

Comment b. Focus on critical insights and avoid restating results already presented.

Response b: We have removed some duplicate results and added unique concerns. Please see L448-L475, L515-L537, 570-L584.

CONCLUSION

Comment a. Include specific recommendations for physical education practices, such as strategies for enhancing teacher support and fostering student engagement in sports.

Response a: We have added a number of descriptions of the findings to extend the practical recommendations of this study for teaching and learning in physical education. Please see L645-L659.

Comment b. Highlight potential areas for future research, such as examining other mediators or exploring interventions in different cultural contexts.

Response b: We add descriptions of potential areas similar to this study. Please see L645-L659.

LIMITATIONS

Comment a. Expand on the limitations related to the study design and sample characteristics.

Response a: We extend the description of limitations related to study design and sample characteristics. Please see L686-L713.

Comment b. Suggest directions for future research, such as incorporating longitudinal designs or exploring additional influencing factors.

Response b: We have added a description of future research directions. Please see L786-L713.

---

## [Decision Letter · Decision Letter 1]

20 Dec 2024

The Relationship Between Physical Education Teachers' Competence Support and Middle School Students' Participation in Sports: A Chain Mediation Model of Perceived Competence and Exercise Persistence

PONE-D-24-49663R1

Dear Dr. Yan,

We’re pleased to inform you that your manuscript has been judged scientifically suitable for publication and will be formally accepted for publication once it meets all outstanding technical requirements.

Kind regards,

Henri Tilga, PhD

Academic Editor

PLOS ONE

Additional Editor Comments (optional):

Reviewers' comments:

Reviewer's Responses to Questions

**Comments to the Author**

1. If the authors have adequately addressed your comments raised in a previous round of review and you feel that this manuscript is now acceptable for publication, you may indicate that here to bypass the “Comments to the Author” section, enter your conflict of interest statement in the “Confidential to Editor” section, and submit your "Accept" recommendation.

Reviewer #1: (No Response)

2. Is the manuscript technically sound, and do the data support the conclusions?

Reviewer #1: (No Response)

3. Has the statistical analysis been performed appropriately and rigorously? 

Reviewer #1: (No Response)

4. Have the authors made all data underlying the findings in their manuscript fully available?

Reviewer #1: (No Response)

5. Is the manuscript presented in an intelligible fashion and written in standard English?

Reviewer #1: (No Response)

6. Review Comments to the Author

Reviewer #1: (No Response)

7. PLOS authors have the option to publish the peer review history of their article (what does this mean?). If published, this will include your full peer review and any attached files.

Reviewer #1: No

---

## [Editor Report · Acceptance letter]

27 Dec 2024

PONE-D-24-49663R1 

PLOS ONE

Dear Dr. Yan, 

I'm pleased to inform you that your manuscript has been deemed suitable for publication in PLOS ONE. Congratulations! Your manuscript is now being handed over to our production team.

Kind regards, 

on behalf of

Dr. Henri Tilga 

Academic Editor

PLOS ONE